# Correlation between novel inflammatory markers and carotid atherosclerosis: A retrospective case-control study

**Man Liao, Lihua Liu, Lijuan Bai, Ruiyun Wang, Yun Liu, Liting Zhang, Jing Han, Yunqiao Li\*, Benling Qi⬤\***

Department of General, Union Hospital, Tongji Medical College, Huazhong University of Science and Technology, Wuhan, Hubei Province, China

\* liyunqiao@hust.edu.cn (YL); qibenlingok_2015@163.com (BQ)

**Data Availability Statement:** All relevant data are within the manuscript and its Supporting information files.

## Abstract

### Objective

Carotid atherosclerosis is a chronic inflammatory disease, which is a major cause of ischemic stroke. The purpose of this study was to analyze the relationship between carotid atherosclerosis and novel inflammatory markers, including platelet to lymphocyte ratio (PLR), neutrophil to lymphocyte ratio (NLR), lymphocyte to monocyte ratio (LMR), platelet to neutrophil ratio (PNR), neutrophil to lymphocyte platelet ratio (NLPR), systemic immune-inflammation index (SII), systemic inflammation response index (SIRI), and aggregate index of systemic inflammation (AISI), in order to find the best inflammatory predictor of carotid atherosclerosis.

### Method

We included 10015 patients who underwent routine physical examinations at the physical examination center of our hospital from January 2016 to December 2019, among whom 1910 were diagnosed with carotid atherosclerosis. The relationship between novel inflammatory markers and carotid atherosclerosis was analyzed by logistic regression, and the effectiveness of each factor in predicting carotid atherosclerosis was evaluated by receiver operating characteristic (ROC) curve and area under the curve (AUC).

### Result

The level of PLR, LMR and PNR in the carotid atherosclerosis group were lower than those in the non-carotid atherosclerosis group, while NLR, NLPR, SII, SIRI and AISI in the carotid atherosclerosis group were significantly higher than those in the non-carotid atherosclerosis group. Logistic regression analysis showed that PLR, NLR, LMR, PNR, NLPR, SII, SIRI, AISI were all correlated with carotid atherosclerosis. The AUC value of NLPR was the highest, which was 0.67, the cut-off value was 0.78, the sensitivity was 65.8%, and the specificity was 57.3%. The prevalence rate of carotid atherosclerosis was 12.4% below the cut-off, 26.6% higher than the cut-off, and the prevalence rate increased by 114.5%.

**Funding:** The study was supported by the National Natural Science Foundation of China (Grant No.81571373, No.81601217, No.82001491), Natural Science Foundation of Hubei Province of China (Grant No. 2017CFB627), Health Commission of Hubei Province scientific research project (Grant No. WJ2021M247) and Scientific Research Fund of Wuhan Union Hospital (Grant No.2019). The funders had no role in study design, data collection and analysis, decision to publish, or preparation of the manuscript.

**Competing interests:** The authors have declared that no competing interests exist.

## Conclusion

New inflammatory markers were significantly correlated with carotid atherosclerosis, among which NLPR was the optimum inflammatory marker to predict the risk of carotid atherosclerosis.

## Introduction

Cardiovascular was always the main cause of premature death and rising healthcare costs [1, 2]. From 1990 to 2019, the total number cardiovascular disease cases has nearly doubled globally, while the number of cases and deaths from peripheral artery disease tripled [3]. Atherosclerosis, a major pathological process in most cardiovascular diseases, which may occuar as early as childhood and remain latent in the body for a long time [4]. Early detection of arterial disease in seemingly healthy individuals focuses on the peripheral arteries, especially the carotid arteries [5]. In clinical practice, carotid atherosclerosis (CAS) is the earlier and most easily detected form of atherosclerosis. Besides, carotid atherosclerotic plaque is an independent risk factor for stroke and coronary heart disease [6]. Atherosclerosis is widely recognized as a chronic inflammatory disease of the blood vessels caused by the accumulation of low density lipoprotein cholesterol [7]. Chronic inflammation is a low-grade, non-infectious, systemic inflammatory state that is associated with age, psychology, environment, lifestyle, and the resolution of acute inflammation [8]. Chronic inflammation is associated with endothelial dysfunction, leukocyte recruitment, transformation of monocytes into macrophages and eventually into foam cells, smooth muscle cell migration and other processes [7]. Chronic inflammation is involved in the whole process of the occurrence and development of atherosclerosis and is the core of atherosclerosis.

In clinical practice, peripheral blood cell count is often used as a predictor and evaluation factor for the severity of inflammation and treatment outcome of acute inflammatory diseases, such as lung infection and sepsis. Platelet to lymphocyte ratio (PLR), neutrophil to lymphocyte ratio (NLR), lymphocyte to monocyte ratio (LMR), platelet to neutrophil ratio (PNR), neutrophil to lymphocyte platelet ratio (NLPR), systemic immune-inflammation index (SII), systemic inflammation response index (SIRI), and aggregate index of systemic inflammation (AISI) are commonly used blood cell count derived values in clinical practice, also known as novel inflammatory markers. These novel inflammatory markers have repeatedly demonstrated their potential value in the early prediction and prognosis of cardiovascular disease [9–14]. However, the relationship between these readily available inflammation markers and atherosclerosis, especially carotid atherosclerosis, has not yet been clear, which nedds further research to be confirmed.

Therefore, this study aimed to explore the correlation between novel inflammatory markers and CAS, in order to find a better early warning indicator of clinical carotid atherosclerosis.

## Method

### Study design

This is a retrospective case-control study. All methods were carried out in accordance with relevant guidelines and regulations and no important aspects of the study have been omitted. Patients' personal information is being kept confidentially. The study complies with the Declaration of Helsinki and the ethical approval of the study was obtained from the Ethics

Committee of Union Hospital, Tongji Medical College, Huazhong University of Science and Technology at Sep. 4th 2023 ([2023]ID:0611). The ethics committee exempted the need for informed consent.

## Population

In this study, we included 14,115 patients who underwent routine physical examinations at our physical examination center from January 2016 to December 2019. The first time we accessed the data was on September 10, 2023. The following were the inclusion criteria for this study: (1) Age>18 years; (2) People who have underwent carotid vascular ultrasound. The exclusion criteria of participants were as follows: (1) Receiving or being receiving anti-inflammatory therapy within 6 months; (2) Critically ill patients with unstable vital signs; (3) Have received or are receiving glucocorticoid therapy within 6 months; (4) Incomplete clinical data or incomplete personal information. After screening, a total of 10015 patients were included and divided into CAS group and non-CAS group according to diagnosis.

## Collected clinical data and laboratory indicators and definition of inflammatory markers

Age, gender, height, and weight of the patient at admission were obtained from the hospital electronic system, and BMI was calculated according to BMI = weight/height ^2. Proper amount of venous blood was extracted by professional nurses with nursing qualification during the fasting period and sent to the laboratory for analysis to obtain white blood cell count (WBC), neutrophil count (NC), lymphocyte count (LC), monocyte count (MC), platelet (PLT), neutrophil percentage (NP), lymphocyte percentage (LP), monocyte percentage (MP), PLR, NLR, LMR, PNR, NLPR, SII, SIRI, AISI and biochemical index, such as aspartate aminotransferase (AST), triglyceride (TG), total cholesterol (TC), high density lipoprotein cholesterol (HDL-C), low density lipoprotein cholesterol (LDL-C), uric acid (UA), urea nitrogen (BUN), creatinine. The estimated glomerular filtration rate (eGFR) was calculated based on the modified MDRD equation. SIRI was defined as neutrophil count*monocyte count/lymphocyte count. SII was calculated using the formula: NLR*platele. AISI was calculated using the formula: (neutrophil count*monocyte count*platelet)/lymphocyte count. NLPR was calculated using the formula: neutrophil count*100/lymphocyte count*platelet. Comorbidity, smoking history and drinking history were obtained from the patient's personal history.

## Diagnostic criteria for carotid atherosclerosis

Carotid atherosclerosis is diagnosed in one of the following situations: (1) A clear history of atherosclerosis or revascularization treatment; (2) Carotid artery ultrasound showed atherosclerotic plaque or carotid intima-media thickness of 1.0mm or more [5].

## Statistical analysis

Shapiro-Wilkstest is used to test whether the continuous variables obey normal distribution, and the continuous variables with normal distribution are represented by mean ± standard deviation (SD). Continuous variables that are not normally distributed are expressed as medians with interquartile range (IQR). The categorical variable is represented by number with percentages (%). ANOVA tests (conforming to normal distribution) and Kruskal-Wallis tests (non-conforming to normal distribution) were used for the differences between of continuous data among CAS and non-CAS. Chi-square test was used to compare the difference of Categorical data between patients with CAS and non-CAS. Logistic regression analysis was used to

evaluate the correlation between various inflammatory indicators and CAS after adjusting factors such as age, gender, BMI, serum biochemical indicators and clinical diagnosis and other factors. In Logistic regression analysis, forward selection method was used to identify CAS risk factors. Finally, the efficiency of each inflammatory markers to CAS was evaluated by receiver operating characteristic (ROC) curve. P<0.05 was considered to be statistically significant. All statistical analyses and diagrams were done using SPSS (version 23.0) or R language.

## Result

### Clinical baseline data

A total of 10015 patients aged 18–94 were enrolled in the study, including 1910 (19.1%) in the CAS group. Table 1 shows the baseline characteristics between the two groups. There were no significant differences in BMI, PLR and dyshepatia between the two groups (P>0.05). While age, gender, WBC, NC, LC, MC, PLT, NP, LP, MP, NLR, LMR, PNR, NLPR, SII, SIRI, AISI, AST, TG, TC, HDL_C, LDL_C, BUN, creatinine, eGFR, UA, smoking history, drinking history, renaldysfuncyion, hyperuricemia, fatty liver, dyslipidemia, hypertension, diabetes and osteoporosis were significantly different between the two groups(P<0.05).

### Relationship between novel inflammatory markers and CAS

The relationship between novel inflammation markers and CAS was observed by spline smoothing plot, shown in Fig 1. LMR, PNR were negatively correlated with CAS, while PLR, NLR, NLPR, SII, SIRI, AISI, NLPR were positively correlated with CAS.

### Logistic regression analysis of CAS

Univariate logistic regression analysis of the relationship between inflammatory markers and CAS showed the same difference as that between the two groups. Furthermore, after adjusting for many possible confounders such as age, sex, BMI, all inflammatory markers were statistically significant (P<0.05) with CAS. These infalmmatory markers have varying effects on CAS, and the risk value of NLPR is much higher than other markers (OR = 2.35). Their CAS risk value and forest plot were shown in Fig 2.

   The efficacy of inflammatory markers to predict CAS was assessed by ROC curve. After adjusting for confounding factors, there were still statistical differences in all inflammatory markers. The ROC curve to evaluate the effectiveness of the above markers in CAS diagnosis is shown in Table 2 and Fig 3. The AUC values of NLR, SII, SIRI, AISI and NLPR were all in the range of 0.5–0.7. The highest AUC value for NLPR recognition of CAS is 0.67. And NLPR uses the adjusted predictive value of Model 3 to identify the AUC value of CAS as 0.91. The cut-off value calculated by Youden index was that NLPR was 0.78, that the sensitivity was 65.8%, and that the specificity was 57.3%. The prevalence rate of carotid atherosclerosis was 12.4% below the cut-off, 26.6% higher than the cut-off, and the prevalence rate increased by 114.5%.

### Subgroup analyses between NLPR and CAS

In order to study the CAS risk and interaction of inflammatory markers in people with different clinicopathological characteristics. We further analyzed six subgroups of NLPR and CAS risk indicators (hyperuricemia, fatty liver, dyslipidemia, hypertension, diabetes, osteoporosis). The results showed that NLPR had the highest OR value in population without lipid metabolism abnormalities (Table 3). CAS were significantly associated with NLPR in all subgroups. The interaction among subgroups revealed that NLPR had an interaction with hyperuricemia,

**Table 1. Baseline characteristics between CAS and non-CAS.**

| | Non-CAS | CAS | P value |
|---|---|---|---|
| N(%) | 8105(80.9) | 1910(19.1) | |
| Age,year | 43.21 ± 12.65 | 66.12 ± 12.07 | 0.000 |
| Male, N(%) | 5105(63.0) | 1605(84.0) | 0.000 |
| BMI, Kg/m2 | 23.80 ± 3.30 | 24.53 ± 3.19 | 0.148 |
| WBC | 6.11 ± 1.46 | 6.29 ± 1.59 | 0.000 |
| NC | 3.50 ± 1.09 | 3.78 ± 1.25 | 0.000 |
| LC | 2.08 ± 0.55 | 1.91 ± 0.56 | 0.000 |
| MC | 0.36 ± 0.12 | 0.39 ± 0.13 | 0.000 |
| PLT | 235.57 ± 54.26 | 212.40 ± 52.94 | 0.000 |
| NP | 56.69 ± 7.44 | 59.51 ± 7.82 | 0.000 |
| LP | 34.48 ± 7.05 | 31.04 ± 7.44 | 0.000 |
| MP | 5.95 ± 1.53 | 6.35 ± 1.67 | 0.000 |
| PLR | 120.07 ± 38.23 | 118.90 ± 41.34 | 0.239 |
| NLR | 1.76 ± 0.61 | 2.12 ± 0.91 | 0.000 |
| LMR | 6.19 ± 2.10 | 5.24 ± 1.93 | 0.000 |
| PNR | 72.54 ± 24.68 | 60.64 ± 20.90 | 0.000 |
| NLPR | 0.78 ± 0.31 | 1.06 ± 0.54 | 0.000 |
| SII | 418.42 ± 188.35 | 452.74 ± 237.00 | 0.000 |
| SIRI | 0.65 ± 0.36 | 0.86 ± 0.56 | 0.000 |
| AISI | 156.07 ± 104.87 | 186.33 ± 137.93 | 0.000 |
| TP, g/L | 74.86 ± 4.12 | 74.35 ± 4.40 | 0.000 |
| Albumin, g/L | 47.27 ± 2.54 | 46.30 ± 2.68 | 0.000 |
| Globulin, g/L | 27.58 ± 3.31 | 28.05 ± 3.77 | 0.000 |
| AGR | 1.74 ± 0.23 | 1.68 ± 0.25 | 0.000 |
| AST, U/L | 25.80 ± 19.48 | 24.48 ± 18.33 | 0.008 |
| TG, mmol/L | 1.58 ± 1.23 | 1.65 ± 1.09 | 0.016 |
| TC, mmol/L | 4.75 ± 0.86 | 4.84 ± 1.01 | 0.000 |
| HDL_C, mmol/L | 1.40 ± 0.34 | 1.37 ± 0.33 | 0.000 |
| LDL_C, mmol/L | 2.74 ± 0.70 | 2.84 ± 0.83 | 0.000 |
| BUN, mmol/L | 4.76 ± 1.21 | 5.65 ± 1.54 | 0.000 |
| Creatinine, umol/L | 71.46 ± 14.61 | 79.76 ± 19.64 | 0.000 |
| eGFR, mL/min/1.73m2 | 109.06 ± 22.75 | 93.81 ± 22.49 | 0.000 |
| Uric acid, mmol/L | 363.82 ± 96.91 | 375.34 ± 88.49 | 0.000 |
| Smoking history,N(%) | 2642(32.6) | 830(43.5) | 0.000 |
| Drinking history,N(%) | 1901(23.5) | 591(30.9) | 0.000 |
| Renaldysfuncyion,N(%) | 31(0.4) | 103(5.4) | 0.000 |
| Dyshepatia,N(%) | 83(1.0) | 13(0.7) | 0.166 |
| Dyslipidemia,N(%) | 5231(64.5) | 1402(73.4) | 0.000 |
| Hyperuricemia,N(%) | 4044(49.9) | 1096(57.4) | 0.000 |
| Fatty liver,N(%) | 2484(30.6) | 718(37.6) | 0.000 |
| Diabetes,N(%) | 339(4.2) | 297(15.5) | 0.000 |
| Osteoporosis,N(%) | 745(9.2) | 446(23.4) | 0.000 |

(*Continued*)

**Table 1.** (Continued)

|  | Non-CAS | CAS | P value |
|---|---|---|---|
| Hypertension,N(%) | 1581(19.5) | 835(43.7) | 0.000 |

Continuous variables are presented as mean ± S.D. Categorical data are presented as number (percentages).

CAS, carotid atherosclerosis; BMI, body mass index; WBC, white blood cell; NC, neutrophil count; LC, lymphocyte count; MC, monocyte count; PLT, platelet; NP, neutrophil percentage; LP, lymphocyte percentage; MP, monocyte percentage; PLR, platelet to lymphocyte ratio; NLR, neutrophil to lymphocyte ratio; LMR, lymphocyte to monocyte ratio; PNR, platelet to neutrophil ratio; NLPR, neutrophil to lymphocyte platelet ratio; SII, systemic immune-inflammation index; SIRI, systemic inflammation response index; AISI, aggregate index of systemic inflammation; TP, total protein; AGR, albumin-globulin ratio; AST, aspartate aminotransferase; TG, triglyceride; TC, total cholesterol; HDL_C, high density lipoprotein cholesterol; LDL_C, low density lipoprotein cholesterol; BUN, blood urea nitrogen; eGFR, estimated glomerular filtration rate.

fatty liver, dyslipidemia, hypertension, diabetes and osteoporosis, and all of these disease sub-groups significantly weakened the risk of NLPR for CAS.

## Discussion

Novel inflammatory markers have been proven to be closely related to various diseases, and their easy availability has important clinical value. In this cross-sectional study, we analyzed the effect of multiple inflammatory markers on CAS, where NLR, SII, SIRI, AISI, NLPR remained significant after adjusting for multiple confounding factors. At the same time, this

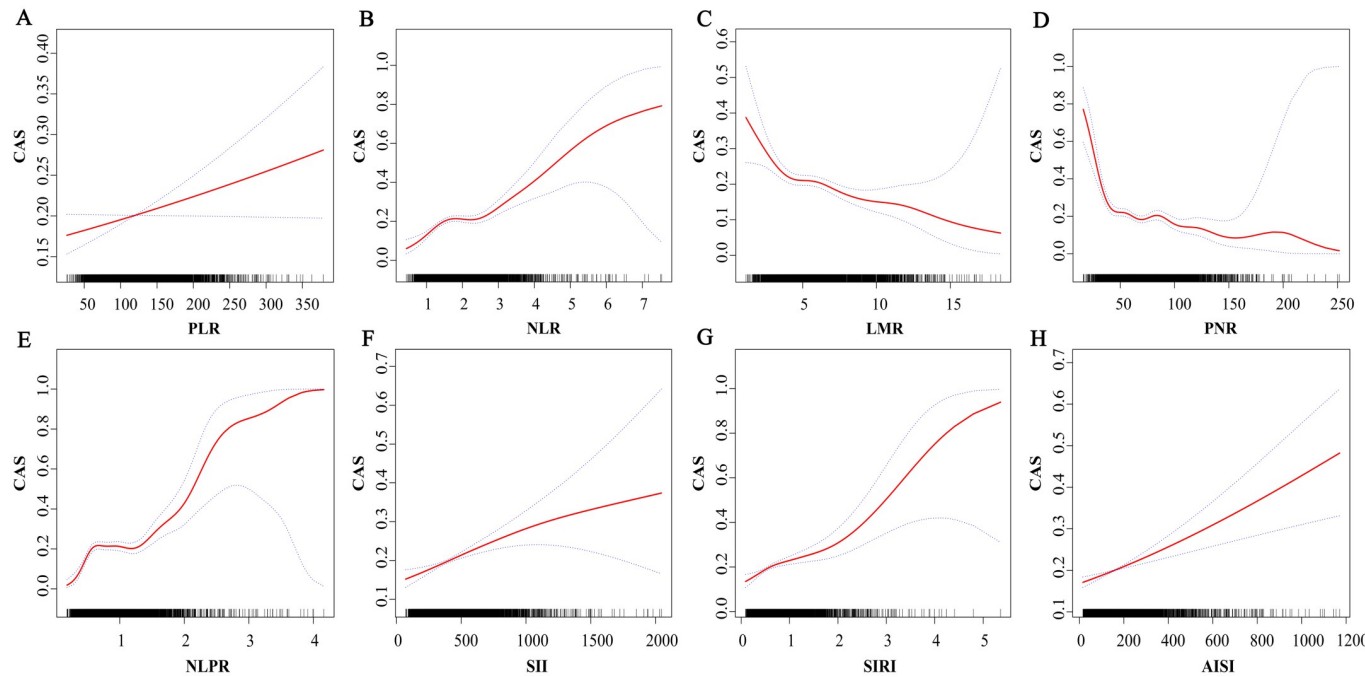

**Fig 1. Spline smoothing plot between novel inflammation markers and carotid atherosclerosis.** A non- linear relationship between novel inflammation markers and carotid atherosclerosis after adjusting for age, gender, body mass index, aspartate aminotransferase, triglyceride, total cholesterol, high density lipoprotein cholesterol, low density lipoprotein cholesterol, estimated glomerular filtration rate, uric acid, hypertension, diabetes, osteoporosis, fatty liver, smoking history, drinking history. CAS, carotid atherosclerosis; PLR, platelet to lymphocyte ratio; NLR, neutrophil to lymphocyte ratio; LMR, lymphocyte to monocyte ratio; PNR, platelet to neutrophil ratio; NLPR, neutrophil to lymphocyte platelet ratio; SII, systemic immune-inflammation index; SIRI, systemic inflammation response index; AISI, aggregate index of systemic inflammation.

| | Model 1 | | Model 2 | | Model 3 | | | |
|---|---|---|---|---|---|---|---|---|
| | OR (95%CI) | Pvalue | OR (95%CI) | Pvalue | OR (95%CI) | Pvalue | OR<1 | OR>1 |
| PLR | 1.00(1.00,1.00) | 0.239 | 1.00(1.00,1.00) | 0.025 | (1.00,1.00) | 0.016 | | |
| NLR | 1.94(1.81,2.08) | 0.000 | 1.48(1.35,1.62) | 0.000 | 1.50(1.37,1.65) | 0.000 | | |
| LMR | 0.77(0.75,0.79) | 0.000 | 0.93(0.90,0.96) | 0.000 | 0.92(0.88,0.95) | 0.000 | | |
| PNR | 0.98(0.97,0.98) | 0.000 | 0.99(0.99,0.99) | 0.000 | 0.99(0.99,0.99) | 0.000 | | |
| NLPR | 5.65(4.94,6.46) | 0.000 | 2.12(1.79,2.52) | 0.000 | 2.35(1.97,2.80) | 0.000 | | |
| SII | 1.00(1.00,1.00) | 0.000 | 1.00(1.00,1.00) | 0.000 | 1.00(1.00,1.00) | 0.028 | | |
| SIRI | 2.86(2.56,3.20) | 0.000 | 1.71(1.47,1.99) | 0.000 | 1.73(1.48,2.01) | 0.000 | | |
| AISI | 1.00(1.00,1.00) | 0.000 | 1.00(1.00,1.00) | 0.000 | 1.00(1.00,1.00) | 0.000 | | |

**Fig 2. The ORs and 95% CI of novel inflammatory indicators to CAS risk by logistic regression.** Model 1 adjusted for none. Model 2 adjusted for age, sex, and body mass index. Model 3 adjusted for age, gender, body mass index, aspartate aminotransferase, triglyceride, total cholesterol, high density lipoprotein cholesterol, low density lipoprotein cholesterol, estimated glomerular filtration rate, uric acid, hypertension, diabetes, osteoporosis, smoking history, drinking history. Forest plot of novel inflammatory markers for carotid atherosclerosis risk after adjusting for Model 3. PLR, platelet to lymphocyte ratio; NLR, neutrophil to lymphocyte ratio; LMR, lymphocyte to monocyte ratio; PNR, platelet to neutrophil ratio; NLPR, neutrophil to lymphocyte platelet ratio; SII, systemic immune-inflammation index; SIRI, systemic inflammation response index; AISI, aggregate index of systemic inflammation.

study further evaluated the value of various inflammatory markers in predicting CAS, among which NLPR had the largest AUC value in identifying CAS risk. Compared with other inflammatory markers, NLPR has a higher value for predicting CAS which may be the best inflammatory indicator for identifying atherosclerosis.

**Table 2. Evaluation of the predictive effect of the novel inflammatory markers on risk of CAS by ROC curves.**

| | AUC (95%CI) | Specificity | Sensitivity | Cut-off | Youden's index |
|---|---|---|---|---|---|
| PLR | 0.52 (0.50,0.53)[a] | 0.76 | 0.28 | 92.41 | 0.04 |
| NLR | 0.62(0.60,0.63)[a] | 0.69 | 0.47 | 1.96 | 0.16 |
| LMR | 0.64(0.62,0.65)[a] | 0.68 | 0.51 | 5.02 | 0.19 |
| PNR | 0.65(0.63,0.66)[a] | 0.64 | 0.57 | 61.24 | 0.21 |
| SII | 0.53(0.52,0.55)[a] | 0.84 | 0.23 | 571.56 | 0.07 |
| SIRI | 0.63(0.62,0.64)[a] | 0.69 | 0.50 | 0.72 | 0.19 |
| AISI | 0.57(0.55,0.58)[a] | 0.66 | 0.45 | 163.70 | 0.11 |
| NLPR | 0.67(0.65,0.68)[a] | 0.57 | 0.66 | 0.78 | 0.23 |
| Model 3-NLPR | 0.91(0.90,0.91)[a] | 0.80 | 0.85 | 0.18 | 0.65 |

CAS, carotid atherosclerosis; ROC, receiver operating curve; AUC, area under the curve; PLR, platelet to lymphocyte ratio; NLR, neutrophil to lymphocyte ratio; LMR, lymphocyte to monocyte ratio; PNR, platelet to neutrophil ratio; NLPR, neutrophil to lymphocyte platelet ratio; SII, systemic immune-inflammation index; SIRI, systemic inflammation response index; AISI, aggregate index of systemic inflammation.

[a]P<0.05.

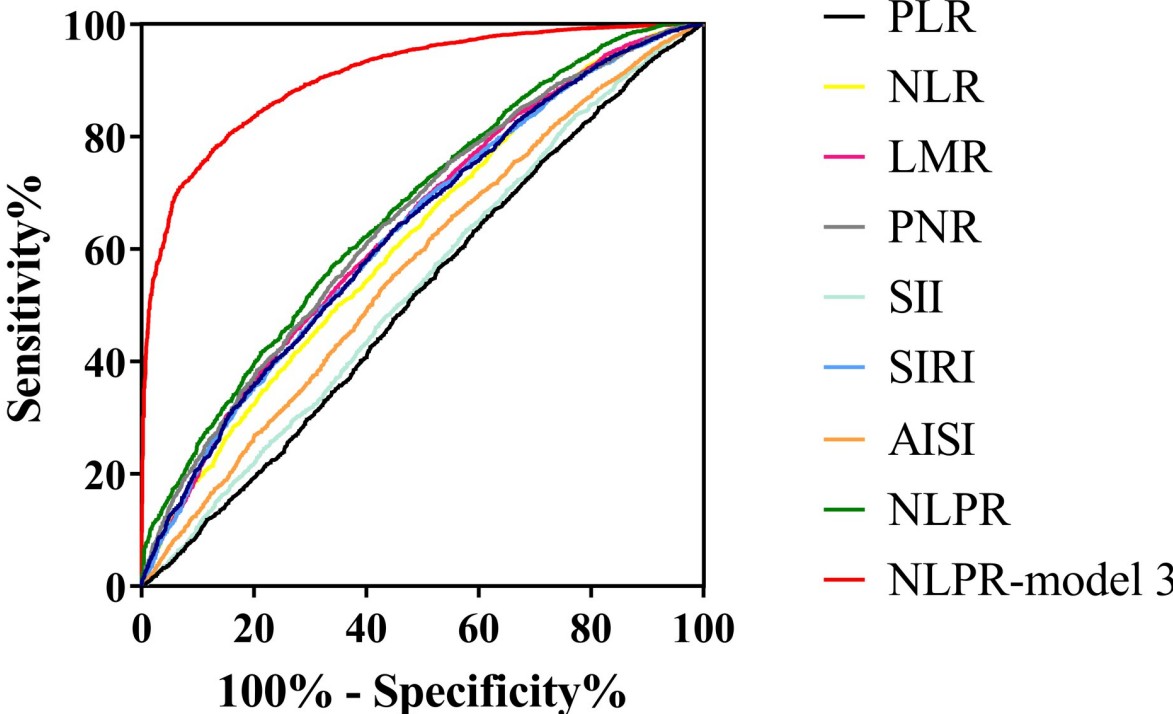

**Fig 3. Receiver operating characteristic (ROC) curves of all inflammatory markers for identifying carotid atherosclerosis risk.** PLR, platelet to lymphocyte ratio; NLR, neutrophil to lymphocyte ratio; LMR, lymphocyte to monocyte ratio; PNR, platelet to neutrophil ratio; NLPR, neutrophil to lymphocyte platelet ratio; SII, systemic immune-inflammation index; SIRI, systemic inflammation response index; AISI, aggregate index of systemic inflammation.

Chronic inflammation leads to atherosclerosis and cardiovascular disease. Neutrophils are the most numerous type of white blood cell and play a major role in inflammation. Although neutrophil count is primarily used as a biomarker for acute infection and inflammation, it has also been shown to accelerate chronic inflammation [15]. Monocyte-derived macrophages are important mediators in the atherosclerotic cascade and are associated with plaque formation through infiltration into the subendothelial layer. Subsequently, uptake of LDL-C complexes leads to the formation of foam cells. In addition to plaque formation, macrophages are involved in extracellular matrix remodeling by secreting pro-inflammatory cytokines and chemokines [16]. In our study, the number of neutrophils and monocytes increased in patients with CAS compared with non-CAS patients, indicating that the higher the number of neutrophils and monocytes, the higher the incidence of CAS. In contrast, lymphocytes slow the progression of atherosclerosis [14]. Our findings were the same, with a reduced number of lymphocytes in patients with CAS, suggesting that the higher the number of lymphocytes, the lower the incidence of CAS. Platelets play two main roles in atherosclerosis: platelets directly adhere to the blood vessel wall to promote plaque formation, and platelets then release inflammatory mediators and chemokines to promote leukocyte recruitment [17]. In our results, platelet counts were reduced in patients with CAS compared to non-CAS patients. The reason for this opposite situation may be that platelets have multiple roles in the formation of atherosclerosis, and in our study, the number of platelets alone was not significantly associated with the development of carotid atherosclerosis. In subsequent binary logistic regression analysis, there was no statistical difference between platelet count and CAS risk after adjusting for confounders.

**Table 3. Association between NLPR and CAS in subgroups.**

|  | N(%) | OR(95%CI) | P interaction |
|---|---|---|---|
| Hyperuricemia |  |  | 0.000 |
| No | 4875 | 2.45(1.89,3.17) |  |
| Yes | 5140 | 2.29(1.81,2.90) |  |
| Fatty liver |  |  | 0.000 |
| No | 6813 | 2.59(2.07,3.25) |  |
| Yes | 3202 | 1.98(1.49,2.63) |  |
| Dyslipidemia |  |  | 0.000 |
| No | 3382 | 2.72(1.89,3.87) |  |
| Yes | 6633 | 2.24(1.83,2.74) |  |
| Hypertension |  |  | 0.000 |
| No | 7599 | 2.56(2.04,3.22) |  |
| Yes | 2416 | 2.14(1.64,2.81) |  |
| Diabetes |  |  | 0.003 |
| No | 9379 | 2.38(1.98,2.87) |  |
| Yes | 636 | 2.26(1.37,3.73) |  |
| Osteoporosis |  |  | 0.000 |
| No | 7755 | 2.55(2.08,3.13) |  |
| Yes | 1191 | 1.91(1.31,2.76) |  |

Above subgroups were adjusted for age, gender, body mass index, aspartate aminotransferase, triglyceride, total cholesterol, high density lipoprotein cholesterol, low density lipoprotein cholesterol, estimated glomerular filtration rate, uric acid, hypertension, diabetes, osteoporosis, smoking history, drinking history.

CAS, carotid atherosclerosis; NLPR, neutrophil to lymphocyte platelet ratio.

In previous studies, we found that NLR, MLR, and PLR increase the risk of arterial stiffness [18]. In recent years, LMR, PNR, NLPR, SII, SIRI, AISI and other indicators have also proved their clinical significance in cardiovascular diseases. A small sample observational study showed that the novel inflammatory markers SIRI, NLR, and LMR were associated with CAS risk in middle-aged and older men [9]. PNR have also been shown to predict mortality in patients with ischemic stroke [10]. SII was also found to be a strong independent predictor of adverse outcomes in patients with acute coronary syndromes [12]. To find the best predictors of CAS, we included these markers in a larger population study at the same time. The results showed that the NLPR had the highest accuracy in identifying the risk of CAS. NLPR is the ratio of neutrophil to lymphocyte*platelet. Initially, NLPR was found to have predictive value in tumor prognosis [19, 20]. Subsequently, NLPR was also found to be associated with the prognosis of acute kidney injury, suppurative liver abscess, severe trauma, and COVID-19 [21–25]. The value of NLPR in cardiovascular disease is equally outstanding. Studies have shown that NLPR is an independent predictor of in-hospital mortality after acute type A aortic dissection [11]. A prospective cohort study found that NLPR was an independent predictor of adverse outcomes in patients with acute coronary syndrome, and patients with a higher NLRP had a higher incidence of major adverse cardiovascular events [26]. In this study, our results directly show that there is a significant correlation between NLPR and CAS, and the higher the level of NLPR, the higher the risk of CAS.

In statistical analyses of baseline characteristics, in addition to indicators of inflammation, we found a number of variables that were statistically different between the CAS and Non-CAS groups, including age, gender, AST, TG, TC, HDL_C, LDL_C, BUN, creatinine, eGFR, UA, smoking history, drinking history, renaldysfuncyion, hyperuricemia, fatty liver,

dyslipidemia, hypertension, diabetes and osteoporosis. Undoubtedly, age is the number one risk factor for a wide range of chronic diseases and systemic chronic inflammatory states [8]. A significant increase in age in the CAS group was also found in our results. Some studies have shown that the incidence of atherosclerosis is higher in postmenopausal women compared to men [27]. Whereas, our results showed higher percentage of males in CAS group, which may be due to the uneven gender distribution in the total population of our medical examination, where males were significantly more than females. In addition to this, renaldysfuncyion, hyperuricaemia, fatty liver, dyslipidaemia, hypertension, diabetes, osteoporosis and history of smoking and alcohol consumption were all strongly associated with atherosclerosis [28–34]. And chronic inflammation can lead to cardiovascular disease, cancer, diabetes, chronic kidney disease, non-alcoholic fatty liver disease and many other diseases [8]. In our subsequent stratified analyses, we also analysed that there was a significant interaction between these diseases and NLPR, which together contributed to the development of CAS.

In this study, we evaluated the value of multiple inflammatory markers for the risk of CAS and found the best inflammatory marker that can be easily obtained to effectively identify the risk of CAS. In addition, the relatively large sample size ensures the accuracy and reliability of the results. Nevertheless, there are some limitations to this study. First, due to the cross-sectional design of the study, a causal relationship between inflammatory markers and CAS cannot be demonstrated, so more prospective studies are needed to confirm these findings. Second, because this study was a single-center study with a relatively narrow group of participants, the findings may not be well extrapolated to other populations. Finally, although we have carefully adjusted for potential confounding factors, it is difficult to rule out potential residual confounding.

## Conclusion

Atherosclerosis is a chronic inflammatory disease of the walls of blood vessels. In this study, novel inflammatory markers have good predictive effects on carotid atherosclerosis, and NLPR has the highest predictive value. NLPR can be used as a potential predictor of carotid atherosclerosis.

## Supporting information

**S1 Raw data.**
(XLSX)

## Author Contributions

**Conceptualization:** Man Liao, Lihua Liu, Yunqiao Li, Benling Qi.

**Formal analysis:** Man Liao, Lihua Liu.

**Funding acquisition:** Lijuan Bai, Ruiyun Wang, Yun Liu, Yunqiao Li, Benling Qi.

**Investigation:** Man Liao, Lihua Liu.

**Supervision:** Lijuan Bai, Ruiyun Wang, Yun Liu, Yunqiao Li, Benling Qi.

**Validation:** Liting Zhang, Jing Han.

**Writing – original draft:** Man Liao, Lihua Liu.

**Writing – review & editing:** Yunqiao Li, Benling Qi.

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
