## [Decision Letter · Decision Letter 0]

20 Feb 2024

PONE-D-23-42287Correlation between Novel Inflammatory Markers and Carotid Atherosclerosis: a Cross-Sectional StudyPLOS ONE

Dear Dr. qi,

Thank you for submitting your manuscript to PLOS ONE. After careful consideration, we feel that it has merit but does not fully meet PLOS ONE’s publication criteria as it currently stands. Therefore, we invite you to submit a revised version of the manuscript that addresses the points raised during the review process.

We look forward to receiving your revised manuscript.

Kind regards,

Elvan Wiyarta, M.D.

Academic Editor

PLOS ONE

Journal Requirements:

"The study was supported by the National Natural Science Foundation of China (Grant No.81571373, No.81601217, No.82001491), Natural Science Foundation of Hubei Province of China (Grant No. 2017CFB627), Health Commission of Hubei Province scientific research project (Grant No. WJ2021M247) and Scientific Research Fund of Wuhan Union Hospital (Grant No.2019)."

3. In this instance it seems there may be acceptable restrictions in place that prevent the public sharing of your minimal data. However, in line with our goal of ensuring long-term data availability to all interested researchers, PLOS’ Data Policy states that authors cannot be the sole named individuals responsible for ensuring data access (http://journals.plos.org/plosone/s/data-availability#loc-acceptable-data-sharing-methods).

**Additional Editor Comments:**

Please address all review from the reviewers below

Please also make the rebuttal letter to address each of the issue

Reviewers' comments:

Reviewer's Responses to Questions

**Comments to the Author**

1. Is the manuscript technically sound, and do the data support the conclusions?

Reviewer #1: Yes

Reviewer #2: Yes

2. Has the statistical analysis been performed appropriately and rigorously? 

Reviewer #1: Yes

Reviewer #2: Yes

3. Have the authors made all data underlying the findings in their manuscript fully available?

Reviewer #1: Yes

Reviewer #2: Yes

4. Is the manuscript presented in an intelligible fashion and written in standard English?

Reviewer #1: Yes

Reviewer #2: Yes

5. Review Comments to the Author

Reviewer #1: Dear Authors, I want to say that this study fulfills the novelty and offers an important result. But, I am concerned about several issues in your manuscript. I think you need to explain more the main rationale of the study and explain why the variable is choosen to investigate. The other comments on the issues have been embedded in the reviewed manuscript. I hope you can revise for the improvement of your manuscript.

line 3 ....In the title the authors mention a cross sectional study design, but in the method section the authors mention the different study design

line 77 ...In the introduction section... Can the authors mention possible factors leading to chronic inflammation which especially is significant in cerebrovascular atherosclerosis?

Line 99... Can authors elaborate why and how significant this variable (CAS) is considered to be investigated in the study. I suggest the explanation can be the rationale of the selection of the CAS in this study?

Line 101..The authors should state consistently the study design

Line 109 ...Can the authors specify the inclusion criteria of subject selection in this study, it can be range of age, gender, and some specific health-related status, etc.

Line 116 ..Do the authors consider the history of TIA, diabetes, smoking history, neck blunt trauma, etc as the confounding factors or exclusion criteria?

Line 120...Please the author provide the timing of lab data collection

Line 141...Is there any temporal relationship between this medical history and the timing of lab data collection?

Line 164..These criteria of subject selection are not yet mentioned in the method section

Line 167 - 172 ...There are several variables which are different between the CAS group and non CAS group. Can the authors interpret these findings?

Line 204 ..Can the authors show the statistical analysis for adjusting these confounding factors?

Reviewer #2: I evaluated this study in which new inflammatory markers were used to predict coronary artery disease. The fact that the number of patients in the study is quite remarkable increases the power of the study. New inflammatory markers were found to be significantly correlated with carotid atherosclerosis; However, NLPR was found to be the most appropriate inflammatory marker to predict the risk of carotid atherosclerosis. These markers may be practical, accessible and inexpensive for the clinician to use in their daily routine. In general, the English language of the well-designed and written study should be reviewed.

6. PLOS authors have the option to publish the peer review history of their article (what does this mean?). If published, this will include your full peer review and any attached files.

Reviewer #1: No

Reviewer #2: No

---

## [Author Response · Author response to Decision Letter 0]

26 Mar 2024

Journal Requirements:

Response: Yes, I've checked that my manuscript meets PLOS ONE's style.

2.Please state what role the funders took in the study.  If the funders had no role, please state: ""The funders had no role in study design, data collection and analysis, decision to publish, or preparation of the manuscript."" If this statement is not correct you must amend it as needed. Please include this amended Role of Funder statement in your cover letter; we will change the online submission form on your behalf.

Response: The funders had no role in study design, data collection and analysis, decision to publish, or preparation of the manuscript. I have affirmed the relevant content in my cover letter.

3.In this instance it seems there may be acceptable restrictions in place that prevent the public sharing of your minimal data. However, in line with our goal of ensuring long-term data availability to all interested researchers, PLOS’ Data Policy states that authors cannot be the sole named individuals responsible for ensuring data access (http://journals.plos.org/plosone/s/data-availability#loc-acceptable-data-sharing-methods).

Response: I'll attach my raw data to the support information.

4.Please include captions for your Supporting Information files at the end of your manuscript, and update any in-text citations to match accordingly. Please see our Supporting Information guidelines for more information: http://journals.plos.org/plosone/s/supporting-information.

Response: All values in the tables and images in the article are based on raw data and statistically analysed. I'll attach my raw data to the support information.

5.Please review your reference list to ensure that it is complete and correct. If you have cited papers that have been retracted, please include the rationale for doing so in the manuscript text, or remove these references and replace them with relevant current references. Any changes to the reference list should be mentioned in the rebuttal letter that accompanies your revised manuscript. If you need to cite a retracted article, indicate the article’s retracted status in the References list and also include a citation and full reference for the retraction notice.

Response: No changes to original references, but we have added some references.

Review Comments to the Author:

line 3 ....In the title the authors mention a cross sectional study design, but in the method section the authors mention the different study design

Response: This study can be described as both cross sectional study and retrospective case-control study, but it prefers retrospective case-control study, so we describe it uniformly as retrospective case-control study both in the title and in the method section.

line 77 ...In the introduction section... Can the authors mention possible factors leading to chronic inflammation which especially is significant in cerebrovascular atherosclerosis?

Response: Possible factors leading to chronic inflammation have been added in the preamble section.

Line 99... Can authors elaborate why and how significant this variable (CAS) is considered to be investigated in the study. I suggest the explanation can be the rationale of the selection of the CAS in this study?

Response: In line 73-76 we add an explanation of why CAS was chosen for the study.

Line 101...The authors should state consistently the study design

Response: We modify study design consistently to retrospective case-control study both in the title and in the method section.

Line 109 ...Can the authors specify the inclusion criteria of subject selection in this study, it can be range of age, gender, and some specific health-related status, etc.

Response: The population included in this study was greater than 18 years age and underwent carotid vascular ultrasound. We have added to the methods section the inclusion criteria in line 117-118.

Line 116 ...Do the authors consider the history of TIA, diabetes, smoking history, neck blunt trauma, etc as the confounding factors or exclusion criteria?

Response: We consider the history of diabetes, hypertension, smoking history, etc as the confounding factors, but the history of TIA and neck blunt trauma were not documented in our medical history and the history of TIA and neck blunt trauma are not a cause of atherosclerosis.

Line 120...Please the author provide the timing of lab data collection

Response: Clinical data and lab data were collected together, and the time of data collection is described on line 115.

Line 141...Is there any temporal relationship between this medical history and the timing of lab data collection?

Response: This study was retrospective. The medical history was recorded on the same day as the laboratory data, and we collected the history and laboratory data uniformly several years after these data existed.

Line 154...Can the authors show the statistical analysis for these confounding factors?

Response: We describe the statistical results of the confounding factors in Table 1 and calculate the differences between the two groups. However, because subsequent studies have focused on novel inflammatory indicators and adding confounding factors may affect the aesthetics of forest plots of novel inflammatory indicator ORs , the analysis of confounding factors was not continued in subsequent logistic regressions.

Line 164...These criteria of subject selection are not yet mentioned in the method section

Response: We have added to the methods section the inclusion criteria in line 117-118.

Line 167 - 172 ...There are several variables which are different between the CAS group and non CAS group. Can the authors interpret these findings?

Response: We have added a relevant explanation of this result in the discussion section in line 130-150.

Line 204...Can the authors show the statistical analysis for adjusting these confounding factors?

Response: We describe the statistical results of the confounding factors in Table 1 and calculate the differences between the two groups. However, because subsequent studies have focused on novel inflammatory indicators and adding confounding factors may affect the aesthetics of forest plots of novel inflammatory indicator ORs , the analysis of confounding factors was not continued in subsequent logistic regressions.

Line 333...The authors can explain in the discussion section about the possible interaction between the inflammatory markers and the known risk factors/comorbid according to the baseline characteristics of the subjects

Response: We have added a relevant explanation of the baseline characteristics and explained the interactions that exist between inflammatory markers and known co-morbidities in the discussion section in line 130-150.

Line 336...The authors need to mention the strength and limitation of the study

Response: In the last paragraph of the discussion section, we describe the strengths and limitations of this study.

---

## [Decision Letter · Decision Letter 1]

2 May 2024

Correlation between Novel Inflammatory Markers and Carotid Atherosclerosis: a Retrospective Case-control Study

PONE-D-23-42287R1

Dear Dr. qi,

We’re pleased to inform you that your manuscript has been judged scientifically suitable for publication and will be formally accepted for publication once it meets all outstanding technical requirements.

Kind regards,

Elvan Wiyarta, M.D.

Academic Editor

PLOS ONE

Additional Editor Comments (optional):

Reviewers' comments:

Reviewer's Responses to Questions

**Comments to the Author**

1. If the authors have adequately addressed your comments raised in a previous round of review and you feel that this manuscript is now acceptable for publication, you may indicate that here to bypass the “Comments to the Author” section, enter your conflict of interest statement in the “Confidential to Editor” section, and submit your "Accept" recommendation.

Reviewer #2: All comments have been addressed

2. Is the manuscript technically sound, and do the data support the conclusions?

Reviewer #2: Yes

3. Has the statistical analysis been performed appropriately and rigorously? 

Reviewer #2: Yes

4. Have the authors made all data underlying the findings in their manuscript fully available?

Reviewer #2: Yes

5. Is the manuscript presented in an intelligible fashion and written in standard English?

Reviewer #2: Yes

6. Review Comments to the Author

Reviewer #2: Ihave reviewed the article. It s a good study. I hope that the results will be beneficial.

thank you

sincerely yours

7. PLOS authors have the option to publish the peer review history of their article (what does this mean?). If published, this will include your full peer review and any attached files.

Reviewer #2: No
